# Measurement and Analysis of the Parameters of Modern Long-Range Thermal Imaging Cameras

**DOI:** 10.3390/s21175700

**Published:** 2021-08-24

**Authors:** Jaroslaw Barela, Krzysztof Firmanty, Mariusz Kastek

**Affiliations:** Institute of Optoelectronics, Military University of Technology, 00-908 Warszawa, Poland; jaroslaw.barela@wat.edu.pl (J.B.); krzysztof.firmanty@wat.edu.pl (K.F.)

**Keywords:** thermal infrared cameras metrology, minimal resolvable temperature difference (MRTD), range detection, recognition and identification (DRI)

## Abstract

Today’s long-range infrared cameras (LRIRC) are used in many systems for the protection of critical infrastructure or national borders. The basic technical parameters of such systems are noise equivalent temperature difference (NETD); minimum resolvable temperature difference (MRTD); and the range of detection, recognition and identification of selected objects (DRI). This paper presents a methodology of the theoretical determination of these parameters on the basis of technical data of LRIRCs. The first part of the paper presents the methods used for the determination of the detection, recognition and identification ranges based on the well-known Johnson criteria. The theoretical backgrounds for both approaches are given, and the laboratory test stand is described together with a brief description of the methodology adopted for the measurements of the selected necessary characteristics of a tested observation system. The measurements were performed in the Accredited Testing Laboratory of the Institute of Optoelectronics of the Military University of Technology (AL IOE MUT), whose activity is based on the ISO/IEC 17025 standard. The measurement results are presented, and the calculated ranges for a selected set of IR cameras are given, obtained on the basis of the Johnson criteria. In the final part of the article, the obtained measurement results are presented together with an analysis of the measurement uncertainty for 10 LRIRCs. The obtained measurement results were compared to the technical parameters presented by the manufacturers.

## 1. Introduction

Long-range observation systems are an essential part of the security system. They are used by the military, police, border guards and rescue services, and they are also an essential element of the strategic infrastructure protection system. They enable the effective observation of an area under night conditions, thus ensuring the protection of the borders and extensive elements of critical infrastructure through the early detection of threats related to a breach of the security system by a potential intruder (Figure 1).

They enable the effective observation of an area under night conditions, thus ensuring the protection of the borders and extensive elements of critical infrastructure through the early detection of threats related to a breach of the security system by a potential intruder (Figure 1). The development of modern long-range cameras based on modern cooled detector arrays, optical systems with variable focal length and advanced image analysis methods, including those based on deep learning methods, has made it possible to achieve ranges of human detection at distances of over 14 km and vehicle detection at distances of over 20 km. DRI ranges of standardised objects are given by camera manufacturers in the description of technical parameters, but they do not always provide the methodology used to calculate these data.

DRI ranges convey the distance at which an observer can detect, recognise and identify a given target (Figure 2). In catalogue data, these data are (most often) presented in graphical or tabular form. The values of these parameters are given for several characteristic targets (human, car, standard NATO target, ship, etc.).

Many factors influence the image we observe on the display of a long-range thermal imaging camera: the parameters of the elements used to build the camera are the monitor, atmospheric conditions, parameters of the target we are looking for, and background parameters (Figure 3). Therefore, real-world tests to determine the parameters of a long-range observational thermal camera, especially the most relevant DRI, are very difficult to perform, and comparing the parameters of several cameras would force them to be collected at the same time in the same place.

We ask the question of whether it is possible to compare the range parameters of surveillance cameras based on the catalogue data of various manufacturers if they do not state how they determined them or which criteria they are based on. This problem becomes particularly important when one has to invest quite considerable financial means in the purchase of tens or sometimes hundreds of expensive observation systems for securing, e.g., a state border. The range parameters of observation systems can be determined based on computer simulations, laboratory measurements or training ground tests [1]. Computer simulations allow theoretical ranges to be determined based on the precise knowledge of the camera construction. Currently, the most professional program for the simulation of range parameters of thermal imaging cameras is NVThermIP, in which calculations of DRI parameters are performed based on the TTP model. The model assumes that an experienced and well-trained observer can detect a target of a given angular size only if the contrast of the object on the device’s screen is greater than the resolving power of the human eye. This means that the model determines the contrast transfer function CTF of the observing system. To do this correctly, the parameters of all components of the observing system must be known in detail. The following parameters must be entered into the software [2,3,4]:Infrared camera system parameters (spectrum cut-off wavelength, magnification, FOV and sensor frame rate);optics (average optical transmission, f-number, MTF and vibration/stabilisation blur-spot size);detector (detector dimension, peak D*, integration time, numbers of detectors, spectral detectivity and fixed pattern noise);electronics (frame integration, interpolation of pixels, e-zoom type and value, type of digital filter);display (type, height, average luminance, spot width, viewing distance, number of eyes used);atmosphere (transmission, smoke);target (contrast, size, V50 search, V50 recognition and V50 identification).

The result of the simulation is a graph of the probability of detection, recognition and identification of the target as a function of distance. However, the use of software requires a large amount of knowledge in the field of thermal imaging and of detailed parameters of the observation system. These parameters cannot be obtained either from catalogue data or from the manufacturer.

Another possibility to determine the DRI range parameter is field-testing. Field tests are expensive and difficult to conduct due to providing the same weather conditions. Figure 3 shows the:visibility;humidity;temperature;atmosphere absorption;turbulence;clouds;sun.

Only by making observations of targets under the same atmospheric conditions is it possible to compare the achieved DRI ranges between individual measurements [2,5,6]. Therefore, it seems reasonable to determine the range parameters based on laboratory measurements.

Under laboratory conditions, more than 20 parameters of thermal imaging cameras can be measured, which determine the quality of the generated image. Manufacturers of observational thermal cameras include over a dozen parameters of the device in their catalogue data, but only two of them are concerned with the observational properties. These are the NETD and DRI ranges (Figure 4). However, we must consider what these data mean. First, very often, manufacturers do not state how these data were determined and for which parameters of the target (dimensions) and environment. Currently, there are at least nine methods for determining DRI parameters [7,8]. The most popular of these are the Johnson criteria—STANAG4349 [9] and STANAG4347 [10]; the FLIR92 model [11]; Targeting Task Performance Metrics TTP [12,13,14,15,16]; and the Thermal Range Model TRM [17]. The DRI results obtained from these vary. The differences are up to 30%. Most often, we do not know the dimensions of the target and the extinction coefficient of the atmosphere. For example, in the literature, we can find three sizes of the human-type target. This means that the comparison of DRI ranges found in the camera catalogue data do not make sense if the measurement conditions and calculation methodology are not the same.

## 2. Materials and Methods

### 2.1. Laboratory Stand for Testing Infrared Cameras

The reliable measurement of parameters of optoelectronic devices can only be performed under laboratory conditions using specialised equipment with known verified parameters and using verified measurement methods. Measurements under field conditions are burdened with a large error as we cannot monitor the parameters of both the target and the atmosphere with sufficient precision. The construction of a laboratory station for measuring the parameters of thermal imaging devices is strictly defined and thoroughly described in the literature [18,19,20,21,22,23,24].

It consists of a collimator, infrared radiation source (blackbody), radiation source controller, rotating wheel with a target test set, and a computer with the measurement card and specialised software. The laboratory stand measurements in Figure 5 should be characterised by parameters ensuring minimal influence on the measurement result [25]. Each component of the test bench must meet strict requirements. In addition, the parameters of the measuring station must not affect the results of the measured camera parameters. We must also have a sufficient number of tests. To correctly determine DRI ranges based on MRTD, it is necessary to have several dozen four-band tests.

There are only a few manufacturers of laboratory stands for thermal camera parameter measurements in the world [26,27,28,29]. This is due to two facts: the very high requirements for the bench components and the few customers. The procedures for measuring the basic parameters of optoelectronic devices are well described in the literature [25,30,31], and some of them are standardised [9,10,32,33].

Laboratory AL IOE MUT has a measuring stand, the selected technical parameters of which are presented in Table 1, which allows for the measurement of more than twenty parameters of observational thermal imaging cameras. In the vast majority of cases, those who order the measurements do not include the measurement of NETD and MRTD and the determination of DRI range parameters. In Europe, the primary means of determining DRI ranges is the Johnson criteria, which are normalised (STANAG4347 and STANAG4349) [9,10].

### 2.2. Parameter NETD

The NETD parameter is one of the parameters describing the noise of thermal imaging systems. The image generated by thermal imaging devices is not a perfect one. There is an unwanted signal in the image called noise. Each of the noise components of the imaging system affects the quality of the generated image. Analysing the influence of individual noise sources on the quality of the generated image is difficult, if not impossible, as several noise types can cause the same image distortion. Therefore, two noise models of imaging systems have been introduced to enable noise analyses to be performed. The 3D model allows eight noise components to be extracted. It is an accurate model used by device designers. The information obtained from it is complete, but of little use to the user of the imaging system. Therefore, the so-called simplified noise model was introduced. It is assumed that noise by type can be divided into temporal and spatial types. In addition, each of these noise types, due to their spectra, can be divided into high-frequency noise and low-frequency noise. Modern thermal imaging systems, using the computational capabilities of processors, eliminate spatial noise (low and high frequency) and low-frequency temporal noise from the image very well. Therefore, the dominant noise source in modern thermal imaging systems is high-frequency temporal noise. High-frequency temporal noise causes fast changes in the signal value of a single pixel in time. This noise causes differences in the signal value of the same pixel in two consecutive images. High-frequency time noise is caused by noise: Johnson–Nyquist, generation–recombination and the shot noise of detectors and electronic components of processing systems. The noise equivalent temperature difference (NETD) is a parameter of thermal imaging devices that depends on this type of noise. The NETD of thermal imaging devices shows the minimum level of the signal reaching the thermal camera, coming from the observed target, that the camera can detect. To determine the high-frequency time-domain noise NETD, measurements of the RMS signal voltage of the spatial noise $U_{n}$, the signal voltage of the signal coming from the test $U_{2}$ with temperature $T_{2}$ and the signal voltage of the signal coming from the background $U_{1}$ with temperature $T_{1}$ should be taken. Both the test and its background should have high emissivity. The difference between the background temperature $T_{1}$ and the test temperature $T_{2}$ should not be more than a few degrees, and it depends on the SiTF characteristics of the thermal imaging camera. The device under test should be positioned to obtain a centrally located image of the test on its screen. The high-frequency time noise NETD is determined from the measurement results according to the following formula:(1)$$NETD=\frac{\left(T_{1}-T_{2}\right)U_{n}}{U_{2}-U_{1}},$$
where $T_{1}$—background temperature; $T_{2}$—test temperature; $U_{n}$—RMS value of noise voltage; $U_{1}$—voltage value of the signal coming from the background; $U_{2}$—voltage value of the signal coming from the test. In the literature, there is another form of the above formula [27,28], using the signal transfer function (SiTF). The measurement of the SiTF function is usually limited to the range of the linear response of the system. The SiTF function is determined from the following relation:(2)$$SiTF=\frac{\left(S_{2}-S_{1}\right)}{\left(T_{2}-T_{1}\right)}.$$

Knowing the function SiTF, NETD can be determined from the following relationship:(3)$$NETD=\frac{U_{n}}{\left(SiTF\right)}.$$

### 2.3. Parameter MRTD

The characteristics of the minimum resolvable temperature difference (MRTD) (ordinate axis [K, C], abscissa [lines/mrad, mrad${}^{-1}$] is defined as the dependence of the minimum temperature difference of the four-bar test and the background temperature, ensuring that all test bars are distinguishable by the observer on the spatial frequency of the test. The procedure for determining the characteristics of the minimum distinguishable temperature difference MRTD of thermal imaging devices is a standardised procedure, STANAG4349 [9]. To determine the MRTD characteristics, a measurement of the minimum temperature difference between the test bars and the background temperature at which the observer can distinguish all four test strips was conducted. The observer can optimise the electronic path gain values, screen brightness and other adjustment mechanisms during the tests within the constraints that exist under real operating conditions, while the observation time is not limited. The tests are carried out first for a positive temperature difference of the test bars against the background temperature and then for a negative difference. The temperature difference at which the observer begins to distinguish all the test bars is determined. The final MRTD values obtained for a single observer are determined according to the following formula:(4)$$MRTD\left(\gamma \right)=\frac{\Delta T_{+}\left(\gamma \right)-\Delta T_{-}\left(\gamma \right)}{2},$$
where $\Delta T_{+}\left(\gamma \right)$—MRTD values are determined for a positive temperature difference between the test bars and the background temperature concerning the moments when the observer begins to distinguish the test bars; $\Delta T_{-}\left(\gamma \right)$—MRTD values determined for a negative temperature difference between the test bars and the background temperature concerning the moments when the observer begins to distinguish the test bars. It is recommended that the measurements be carried out by three observers without visual impairment. The final test results are presented as an average value obtained for all observers participating in the measurements.

### 2.4. Uncertainty Analysis NETD and MRTD

Measurement uncertainty analysis in the AL IOE MUT is performed in accordance with the recommendations of the Central Office of Measures of the Republic of Poland [34] considering the guidelines of measurement standards and the specificity of measurement methods [35,36]. Every measurement made is only accurate to a certain extent. This limitation is due to the imprecision of the instruments used during measurement and the finite precision of the observer’s sense organs. This means that every measurement, even the most precise one, is subject to measurement uncertainty, and the result is an approximation of the true value. According to the standard [36], when estimating the uncertainty of measurement, all uncertainty components that are relevant to the measurement situation should be considered, using an appropriate methods of analysis. It is, therefore, necessary to formulate a mathematical model of the measurement process, in which the input quantities should represent all relevant sources of error. Using the NETD measurement procedure, the result of the measurement is several values of temperature difference. From the results, the Type *A* and Type *B* uncertainty of measurement can be calculated. The Type *A* uncertainty of measurement is calculated as the standard deviation of the arithmetic mean of the measurement results:(5)$$u_{B}\left(NETD\right)=\frac{\sqrt{\frac{1}{n-1}\sum_{i=1}^{n}\left(NETD-\bar{NETD}\right)}}{\sqrt{n}}$$
where *n*—the number of measurements taken; and $\bar{NETD}$—the average value of NETD measurement results.

Type *B* measurement uncertainty is related to the limitation of the measurement apparatus. To measure NETD, we used an infrared standard source, a collimator and a measurement card. Since NETD, in terms of linear processing, does not depend on the temperature difference between the test and the environment, we have two sources of error: the radiation standard and the measurement card. In the case of the radiation standards, the measurement is affected by the accuracy of the temperature setting, the uniformity of the surface and the short-term stability, since the measurement lasts no longer than 20 s. The measurement card introduces greater measurement error. In the available measuring stations [26,27,28,29], we have measuring cards with resolution varying from 8-bit to 16-bit. The limit error of the measurement card is 1 LSB, i.e., the accuracy of the measurement largely depends on the resolution of the measurement card. The uncertainty of measurement type *B* is determined from the following relation:(6)$$u_{B}\left(NETD\right)=\sqrt{{\left(\frac{\delta S_{max}}{S_{n}\sqrt{3}}\right)}^{2}+{\left(\frac{\delta S_{max}}{S_{2}\sqrt{3}}\right)}^{2}+{\left(\frac{\delta S_{max}}{S_{1}\sqrt{3}}\right)}^{2}+{\left(\frac{\delta S_{max}}{T_{2}\sqrt{3}}\right)}^{2}+{\left(\frac{\delta S_{max}}{T_{1}\sqrt{3}}\right)}^{2}},$$
where $S_{1}$, $S_{2}$—average signal voltage value for infrared source temperature; $T_{1}$, $T_{2}$—set temperature of infrared source at the time of measurement; $\delta S_{max}$—limit error of measuring card specified by manufacturer; $T_{1}$—radiation pattern error for temperature; $T_{1}$, $T_{2}$—radiation pattern error for temperature; and $T_{2}$, *k*—coverage factor (*k* = 2).

The uncertainty of measurement NETD is calculated from the following relationship:(7)$$u_{cp}\left(NETD\right)=\sqrt{u_{A}^{2}\left(NETD\right)+u_{B}^{2}\left(NETD\right)}.$$

Considering the dilution factor, the expanded uncertainty is calculated from the following relationship:(8)$$U\left(NETD\right)=k\sqrt{u_{A}^{2}\left(NETD\right)+u_{B}^{2}\left(NETD\right)},$$
where *k*—coverage factor (k=2).

The final result is given as the NETD value and the calculated expanded uncertainty. The analysis of MRTD measurement results is more difficult as more factors influence the measurement result. Two approaches can be found in the literature. The first simplifies the measurement process by assuming that the observer is the main contributor to the error in MRTD measurements. The second analysis of measurement uncertainty considers all factors influencing the measurement result [25]. In the AL IOE MUT, the second way of measurement uncertainty analysis was chosen. To determine the uncertainty of the measurement result, it is necessary to formulate a mathematical model of the measurement process, in which the input quantities should represent all significant sources of error. In the process of measuring the MRTD characteristic, we have the following sources of error:inaccuracy of the temperature stabilisation of the radiation source;temperature drift of the radiation source;non-uniformity of the radiation source;the standard deviation of the MRTD measured by the observers;collimator transmission coefficient error;change in ambient temperature.

The inaccuracy of temperature stabilisation in the infrared source is due to the inaccuracy of the temperature measurement over the entire measurement period of a single MRTD test. Since we are measuring positive and negative values of the temperature for which the observer can detect the test, this measurement takes several minutes. The value of this uncertainty can be determined from the following relationship:(9)$$u\left(\Delta T_{BBA}\right)=\frac{2}{3}\tau \Delta t_{A},$$
where τ—transmission coefficient of the infrared collimator; and $\Delta t_{A}$—temperature reading error of the radiation source.

Source temperature drift can have a large influence on the measurement result. In normal operation, the temperature drift of the source depends on its long-term temperature stability. Therefore, the uncertainty caused by the temperature drift of the IR source can be determined from the following relationship:(10)$$u\left(\Delta T_{BBD}\right)=\frac{2}{3}\tau \Delta t_{s},$$
where $\Delta t_{s}$—temperature drift of the radiation source (long-term stability of the source).

Calculations show that its influence on the total uncertainty of the measurement is not large, but two undesirable phenomena may occur. The first is that temperature fluctuations on the test surface cause the observer to detect the test more easily, which is incorrect from a metrological point of view. The second is the calibration of the temperature measurement. Today’s most accurate temperature measurement systems are based on accurate PT100 or PT1000 sensors, precise reference sources and A/D converters. To enable accurate signal processing, A/D converters periodically perform self-calibration. The system user has no influence over when this occurs. If this process occurs during the measurement of the MRTD value of a specific test, the influence of this phenomenon has an effect on the temperature drift of the reference. This effect is in no way included in the catalogue data of the radiation sources.

The uncertainty due to the temperature inhomogeneity of an IR source can be determined from the following relationship:(11)$$u\left(\Delta T_{BBU}\right)=\frac{2}{3}\tau \Delta t_{u},$$
where $\Delta t_{u}$—temperature inhomogeneity of the radiation pattern.

The inhomogeneity of a radiation source is determined by measuring the homogeneity on 80% of its surface, and this value is included in the catalogue data. It is measured at a temperature difference of 1 ${}^{\circ }$C between the source surface and ambient. During a real measurement, the test is located at the centre of the surface, and it occupies a small part of it. Therefore, the actual effect on the measurement result is sometimes smaller.

The method for determining the MRTD characteristics involves determining the MRTD value as the product of the transmission coefficient of the collimator and the average of the sum of the differences of the recorded positive and negative temperature differences T for the three observers:(12)$$MRTD_{obs}=\tau \frac{\left(\frac{\Delta T_{-}^{1}-\Delta T_{+}^{1}}{2}\right)+\left(\frac{\Delta T_{-}^{2}-\Delta T_{+}^{2}}{2}\right)+\left(\frac{\Delta T_{-}^{3}-\Delta T_{+}^{3}}{2}\right)}{3},$$
where $\Delta T_{+}$—the positive temperature difference at which the observer begins to distinguish the test bars; $\Delta T_{-}$—the negative temperature difference at which the observer begins to distinguish the test bars. The above formula that can be converted to the following form:(13)$$u\left(MRTD_{obs}\right)=\frac{\sqrt{\frac{1}{n-1}\sum_{k=1}^{n}{\left(\Delta T_{k}-\Delta \bar{T}\right)}^{2}}}{\sqrt{n}},$$
where $\Delta T_{+}$—the positive temperature difference at which the observer begins to distinguish the test bars; $\Delta T_{-}$—the negative temperature difference at which the observer begins to distinguish the test bars; *n*—the number of measurements taken; $\Delta \bar{T}$—the average value of the temperature difference results for which we discriminate the test; $\Delta T_{k}$—the value of the temperature difference for which a given observer can discriminate the test.

When measuring MRTD, the experience of the observer is critical. The criteria assume that the observer should see at least 75% of the test area for 50% of the observation time. The criteria are precisely described, but the difference in results between an experienced and an inexperienced observer can be large, especially when measuring the performance of uncooled thermal cameras. Another two sources of error, the change in ambient temperature and the error in the transmission coefficient of the collimator, due to the measurements being made under laboratory conditions, can be ignored. In summary, the total uncertainty of the measurement of MRTD using the Formulas (Equation 9)–(Equation 11) and (Equation 13) can be written in the following form:(14)$$u\left(MRTD\right)=\sqrt{u_{\Delta T_{BBA}}^{2}+u_{\Delta T_{BBD}}^{2}+u_{\Delta T_{BBU}}^{2}+u_{MRTD_{obs}}^{2}}.$$

### 2.5. Determination of the Ranges DRI

The range parameters of observation devices are the most important parameters of observation systems for the user. There are several methods for determining the detection, recognition and target identification range parameters of observation devices. The oldest and most widely used method is the Johnson criteria [18]. The original Johnson criteria were published in 1958, which introduced the concept of detection, orientation, recognition and identification ranges. Detecting a target from a given distance means determining whether something is in the field of view of the observing device. The orientation range allows us to determine whether a target is symmetrical. Target recognition means being able to tell whether it is a tank or a car. Target identification allows us to determine, for example, the type of tank. Johnson determined from range tests how many cycles of the MRTD test, a factor of N50 must occur on a target to determine DRI ranges with a probability of 50. The method of determining DRI from measured MRTD characteristics is described in STANAG4347 [10]. According to STANAG434 [10], to determine detection, recognition and identification ranges, it is necessary to:measure or obtain the MRTD characteristics of the device under test from catalogue data;convert the MRTD characteristics into target detection, recognition and identification characteristics, considering the dimensions of the target, according to the following relations:
(15)$$R_{det}=D\vartheta ;$$
(16)$$R_{rec}=\frac{D}{3}\vartheta ;$$
(17)$$R_{id}=\frac{D}{6}\vartheta ;$$
where *D*—the mean linear dimension of the target; ϑ—the spatial frequency of the measured value of the MRTD characteristic;determine the transmission characteristics of the temperature difference of the target relative to the environment in the atmosphere according to the relation (Beer’s law):
(18)$$\Delta T=\Delta T_{o}e^{\sigma R};$$plot the atmospheric transmission characteristics and the detection, recognition and identification characteristics of the device under test on a single graph;define the detection range as the point of intersection of an atmospheric transmission curve with the device detection curve, the recognition range as the point of intersection of the atmospheric transmission curve with the device recognition curve and the identification range as the point of intersection of the atmospheric transmission curve with the device identification curve.

The range of detection, recognition and identification in the AL IOE MUT is determined from the measured characteristic MRTD according to STANAG4347 [10]. Therefore, the uncertainty of the range measurement depends on the measurement uncertainty of the MRTD characteristic and the number of measurement points and their distribution throughout the measurement range of the MRTD characteristic. To determine the measurement uncertainty of the range parameters of thermal imaging devices resulting from the uncertainty of determining the MRTD characteristic, it is necessary to determine the points of intersection of the atmosphere transmission curve with the detection, recognition and identification curves and to consider the measurement uncertainty of the measured MRTD characteristic. The determined measurement uncertainty was, depending on the thermal imaging camera tested, 5 m. A much larger uncertainty error in the determination of the range parameters was obtained due to the limited number of measurement points available. No laboratory has an infinite number of MRTD tests. Therefore, based on the literature and the manufacturer’s data, it can be concluded that the detection, recognition and identification ranges of thermal imaging devices are given with an accuracy not greater than ±50 m.

### 2.6. Long-Range Infrared Camera

As part of the ongoing measurements of the parameters of long-range thermal imaging cameras in the AL IOE MUT, the NETD and MRTD parameters were measured, and the DRI range parameters were determined for more than 10 thermal imaging cameras with the same technical parameters declared by the manufacturer. Table 2 presents the basic technical parameters of this type of camera.

Performing measurements of the NETD and MRTD parameters, as well as the determination of the DRI range parameters, for so many cameras of the same type allowed us to analyse the obtained parameters in terms of the repeatability and dispersion of the obtained values for individual parameters, as well as to compare them with the parameters declared by the manufacturer.

## 3. Results

### 3.1. Results of the NETD Measurement Measurement

According to the methodologies and measurement procedures used in AL IOE MUT, the measurement of the NETD parameter was performed for 10 LRIRCs of the same type and manufacturer. Table 3 presents the obtained results of NETD measurements and the results of uncertainty analysis according to the methodology presented in Section 2.4 for camera C1.

The measurement stand used in AL IOE MUT has two measurement cards: analogue and digital. The NETD results presented here were measured using the 12-bit analogue measurement card. The measurement areas of the signal coming from the standard and the environment were set to maximally cover the field of view of the cameras. The temperature value of the radiation pattern was optimally chosen for the linear processing range of the cameras. The obtained NETD results presented in Table 4 are close to the data reported by the manufacturer.

### 3.2. Results of the MRTD Measurement

In accordance with the methodology used in AL IOE MUT, directly resulting from STANAG4349 [10], the MRTD parameter was measured for 10 cameras of the same type. In accordance with the methodology, observations of the four-bar test were conducted by three trained observers experienced in conducting such measurements. The results of the measurements were archived in the laboratory, and an example of the measurement results for camera C1 is presented in Table 5. The obtained results allowed us to plot the MRTD characteristics, as shown in Figure 6.

After the measurements were made and the MRTD values determined for each spatial frequency corresponding to a given four-bar test, an uncertainty analysis of the measurements was carried out; sample measurement results are shown in Table 6.

The measurements for 10 cameras of the same type were performed in the same way according to the adopted methodology, and the obtained results were analysed in terms of error analysis. Table 7 presents the obtained results of the determination of MRTD values for each spatial frequency for the tested cameras.

From the results of the MRTD measurements for the 10 cameras presented on Figure 7, one can notice a rather large dispersion in the obtained values, particularly for the higher spatial frequencies, γ = 21.27, γ = 23.27 and γ = 24.82, as well as the impossibility of distinguishing a four-bar test with spatial frequency γ = 25.00 by cameras C1, C2, C3 and C9. The obtained MRTD values for these spatial frequencies directly influences the determined value of the recognition and identification range.

### 3.3. Results of the DRI Calculation

Following the methodology presented in Section 2.5, the DRI parameters were determined for each of the 10 cameras tested. Figure 8 presents a plot of the atmospheric transmission characteristics together with the plots of the relationships (Equation 14)–(Equation 16), which allowed the DRI parameters to be determined.

Similarly, the DRI range values were determined for each of the 10 tested cameras, and these results are presented in Table 8 and Table 9, which also include the DRI parameters that are declared by the manufacturer for this type of camera. Calculations were carried out for two different targets: standard target NATO 2.3 m × 2.3 m and man 0.6 m × 1.6 m, assuming a temperature difference between the object and the background ΔT = 2 K, and an atmosphere attenuation coefficient of 0.2.

Figure 9, Figure 10 and Figure 11 show the detection, recognition and identification ranges for the target (standard NATO target 2.3 m × 2.3 m, ΔT = 2 K) for 10 tested LRIRCs.

If we analyse the results presented above the DRI determination for the target, standard target NATO 2.3 m × 2.3 m, ΔT = 2 K and an atmosphere attenuation coefficient of 0.2, then the large scatter in the determined value of the detection range for the tested cameras can be observed. Moreover, if we compare the obtained results with the value of the detection range declared by the manufacturer, it turns out that none of the tested cameras reached the detection range declared by the manufacturer.

Figure 12, Figure 13 and Figure 14 show the detection, recognition and identification ranges for the target, man 0.6 m × 1.6 m, ΔT = 2 K, for 10 tested LRIRCs.

If we analyse the results presented above of DRI determination for the target: standard target, man 0.6 m × 1.6 m, ΔT = 2 K and an atmosphere attenuation coefficient of 0.2, then the large scatter in the tested cameras for the determined value of the detection range can be observed. The individual figures presented above show the DRI range declared by the camera manufacturer, but none of the tested cameras achieved the declared D range value.

## 4. Discussion

Analysing the obtained results of MRTD measurements and then determining the DRI ranges, one can notice large discrepancies in the obtained parameters of detection, recognition and identification ranges.

If we look at the obtained results for the standard NATO target: 2.3 m × 2.3 m and for the target: 0.6 m × 1.6 mm, with the temperature contrast ΔT = 2 K and an atmosphere transmission attenuation of 0.2 in Table 10, we can see the data for the camera that obtained the worst results and for the camera that obtained the best results for DRI ranges.

The most significant difference can be observed for the NATO target standard 2.3 m × 2.3 m with the detection range between the two cameras reaching up to 1000 m; for the reconnaissance range, 650 m; and for the identification range, up to 700 m. These discrepancies are very significant and from the point of view of the LRIRC operator, may cause great concern regarding not achieving the assumed parameters under real conditions. Similar differences can be observed for the target man: 0.6 m × 1.6 m when determining the detection range between the two cameras, reaching up to 950 m. The results determined in AL IOE MUT during the realised measurements differ to a large extent from the DRI ranges declared by the manufacturer, in particular, the D detection ranges.

Figure 15 and Figure 16 show the summary plots of the 10 camera ranges for both targets at the same temperature contrast ΔT = 2 K and an atmosphere attenuation coefficient of 0.2.

Analysing the presented graphs in Figure 15 and Figure 16, it is possible to observe differences between the different tested cameras, particularly if we compare the obtained results of the measurement of the parameter D relating to the declared parameter of the range D by the manufacturer. Table 11 presents the obtained results of the range parameter D for all cameras for both considered targets, standard NATO target: 2.3 m × 2.3 m and for target: 0.6 m × 1.6 m , with temperature contrast ΔT = 2 K and an atmosphere transmission attenuation of 0.2 regarding the declared range parameter D by the manufacturer. The differences between these parameters reaching up to 2100 m for the NATO target and 1950 m for the man are large discrepancies.

## 5. Conclusions

This paper presents methodologies for the measurement and determination of selected parameters of LRIRCs. An approach to the determination of the most important technical parameter of LRIRCs, i.e., the determination of DRI ranges, is presented. The presented methodologies are implemented and used to verify the parameters of thermal cameras in AL IOE MUT. The verification of parameters through standardised measurements performed in independent measurement laboratories allows for the unbiased verification of selected technical parameters determined by the manufacturer, which are impossible to verify by the end user if they do not have the means in their laboratory. The use of standardised methods for measuring the most important parameters of thermal imaging cameras allows manufacturers to verify the declared parameters of thermal imaging cameras. Performing such measurements in accordance with known standards and norms in independent measurement laboratories allows manufacturers to control and compare different types of cameras from different manufacturers. This way, the end user can choose the LRIRC with the best parameters for their needs. Performing measurements of the DRI parameter under laboratory conditions with a defined and known methodology guarantees the high quality and repeatability of the measurements performed, as presented in Section 2.2, Section 2.3 and Section 2.4, related to measurement error analysis. Independent measurement laboratories are subject to audit procedures in accordance with accepted quality standards, e.g., SO/IEC Guide 58:1993 [35] and ISO/IEC GUIDE 98-3:2008 [36], which ensures the full independence of such laboratories and the high quality of the measurements performed. From the presented results of the measurements of 10 LRIRCs of the same type and from the same manufacturer, with the same parameters declared by the manufacturer, one can see large discrepancies between the individual units of the tested cameras. These discrepancies were not verified at the production stage, and the manufacturer always declared the same technical parameters of the cameras. Discrepancies reaching up to 5% for the detection range parameter (target: standard NATO target 2.3 m × 2.3 m, ΔT = 2 K) constitute quite a large discrepancy between the tested cameras. Discrepancies between the parameters declared by the manufacturer and the measurement results obtained may be due to several reasons:different methodology of determining DRI ranges used by the manufacturer (unfortunately not disclosed by the manufacturer);assembly errors resulting from the combination of individual components of the LRIRC, in particular, the two most important elements: the cooled detector array with the lens optics;the use of various image processing methods (methods of improving image quality) that affect the obtained DRI results;lack of complete control of each LRIRC during the production process by designating DRI ranges for each camera.

Of course, there may be even more reasons for the discrepancy, with those listed above having the greatest impact on the DRI parameter during the implementation of the manufacturing process.

A difference of 9% from the catalogue value for the detection range D may not be a cause for concern at first glance, but when translated into detection range values, the difference reaches up to 2100 m. Such a difference should be a cause for concern when installing this type of camera in systems for the protection of critical infrastructure or borders.

One of the basic conclusions resulting from the analysis presented in this article for the end user of LRIRC systems should be that the parameters of thermal imaging LRIRC should be measured in independent measurement laboratories to verify the parameters specified by the manufacturer.

## Figures and Tables

**Figure 1 sensors-21-05700-f001:**
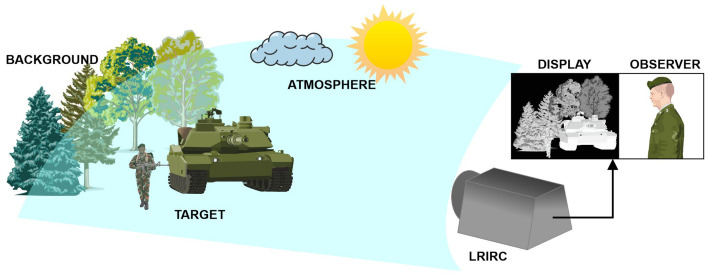
The detection of a target realised by the system.

**Figure 2 sensors-21-05700-f002:**
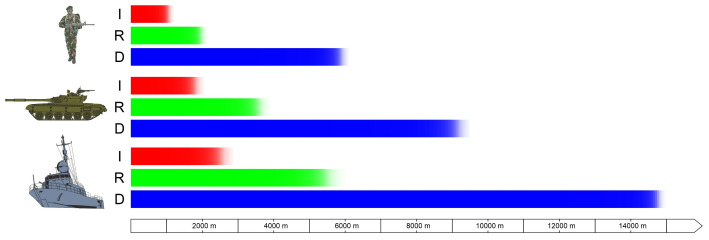
Example of the graphical presentation of the DRI by the producer of the long-range thermal imaging camera.

**Figure 3 sensors-21-05700-f003:**
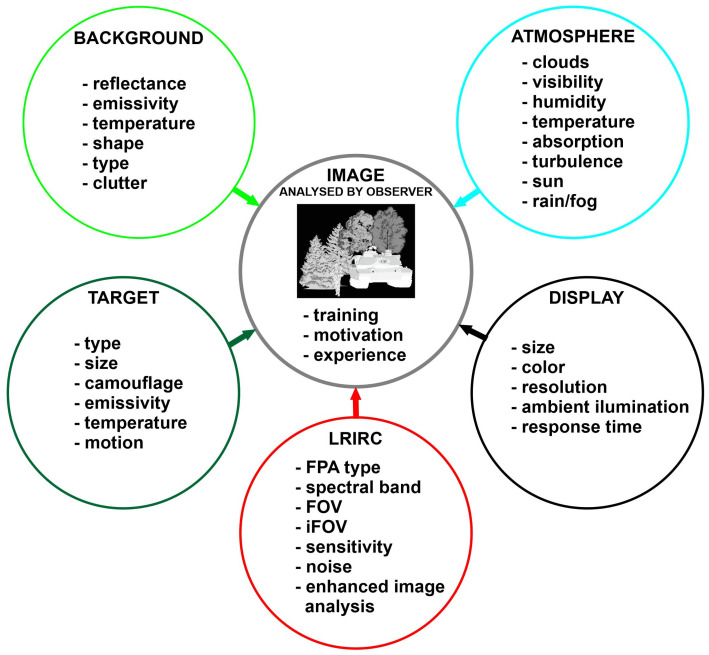
Thermal image forming and perception influence factors.

**Figure 4 sensors-21-05700-f004:**
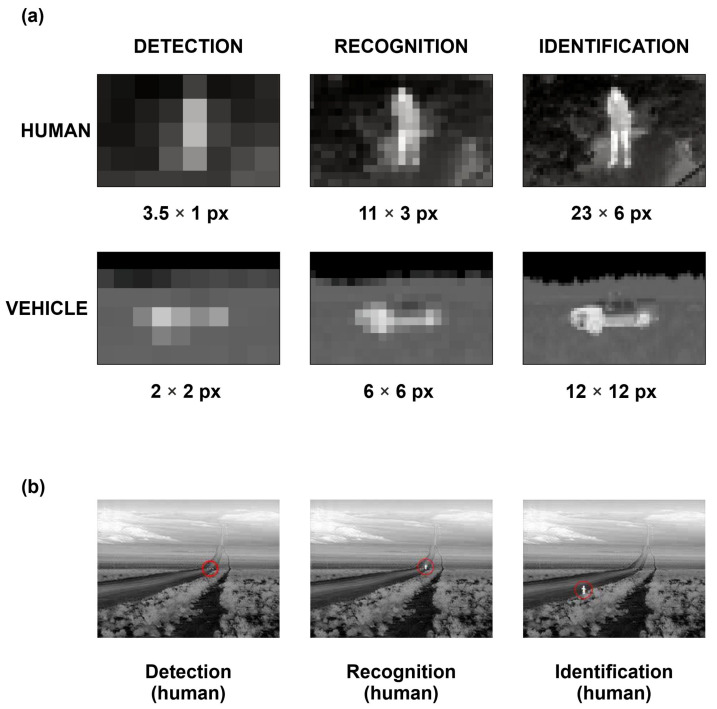
Target detection, recognition and identification: (**a**) illustration of the pixel number on the target necessary for DRI; (**b**) illustration of the target on the real image in ranges for DRI.

**Figure 5 sensors-21-05700-f005:**
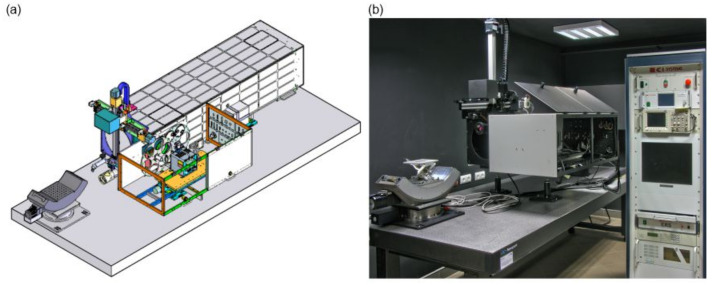
The laboratory stand for infrared cameras: (**a**) the scheme of the stand; and (**b**) the photo of the stand at the AL IOE MUT.

**Figure 6 sensors-21-05700-f006:**
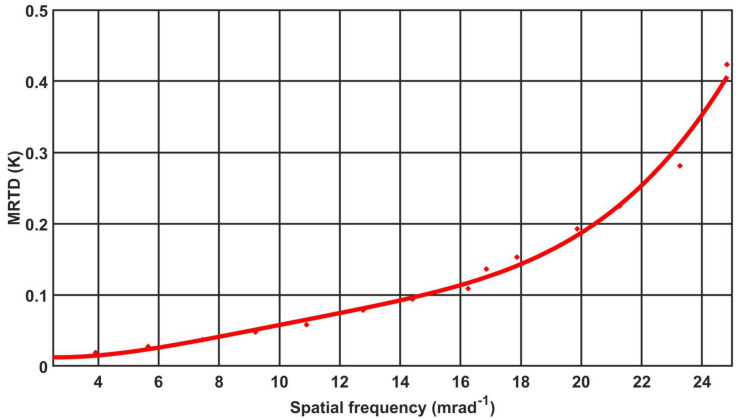
MRTD characteristics measured by camera C1.

**Figure 7 sensors-21-05700-f007:**
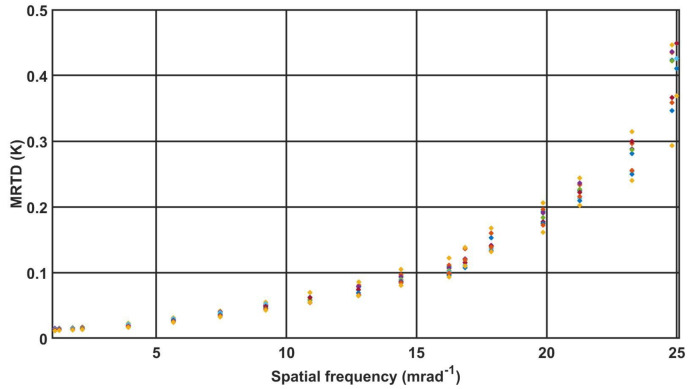
MRTD characteristic 10 cameras.

**Figure 8 sensors-21-05700-f008:**
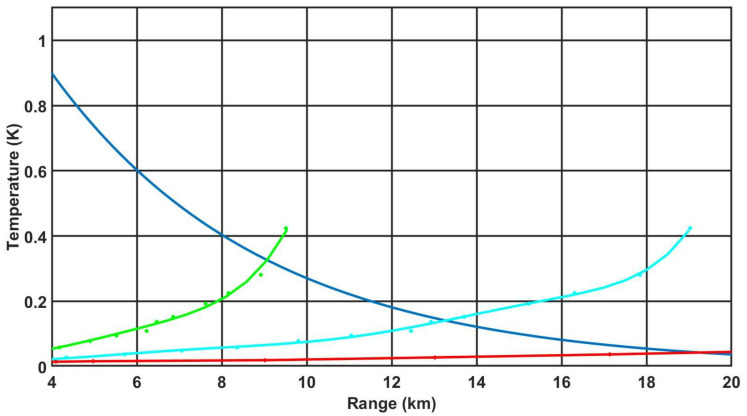
Determination of the DRI for the camera C1. The blue line—the transmission characteristics of the temperature difference of the target relative to the environment in the atmosphere according to the relation (Beer’s law) (Equation 18); the red line—convert the MRTD characteristics into target detection (Equation 15); the blue line—convert the MRTD characteristics into target recognition (Equation 16); and the green line—convert the MRTD characteristics into target detection (Equation 17).

**Figure 9 sensors-21-05700-f009:**
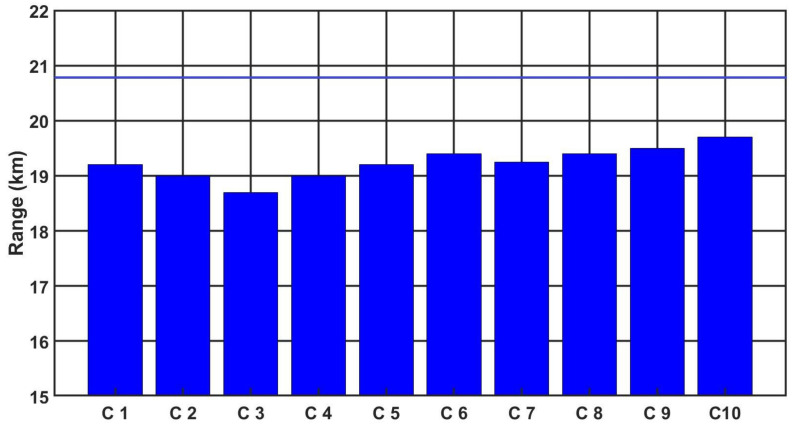
Detection range of the LRIRCs (target: standard target NATO 2.3 m × 2.3 m, ΔT = 2 K). The blue column presents the measured range D for the cameras; the blue line presents the producer declaration of the range D.

**Figure 10 sensors-21-05700-f010:**
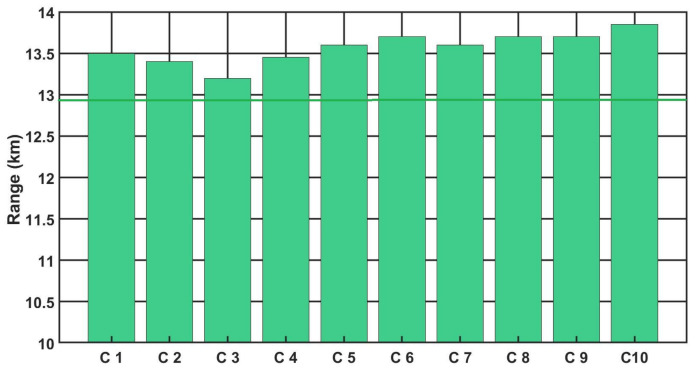
Recognition range of the LRIRCs (target: standard target NATO 2.3 m × 2.3 m, ΔT = 2 K). The green column presents the measured range R for the cameras; the green line presents the producer declaration of the range R.

**Figure 11 sensors-21-05700-f011:**
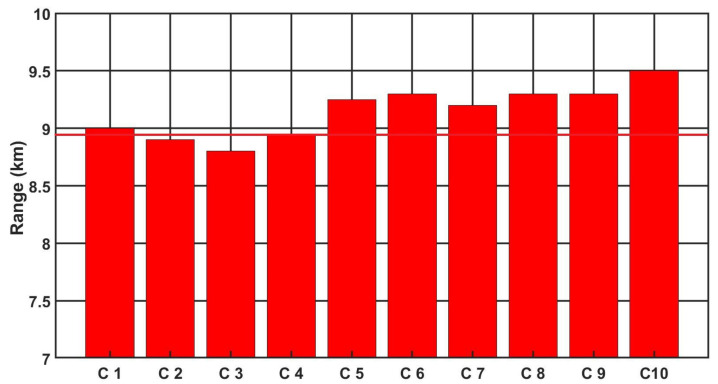
Identification range of the LRIRCs (target: standard target NATO 2.3 m × 2.3 m, ΔT = 2 K). The red column presents the measured range I for the cameras; the red line presents the producer declaration of the range I.

**Figure 12 sensors-21-05700-f012:**
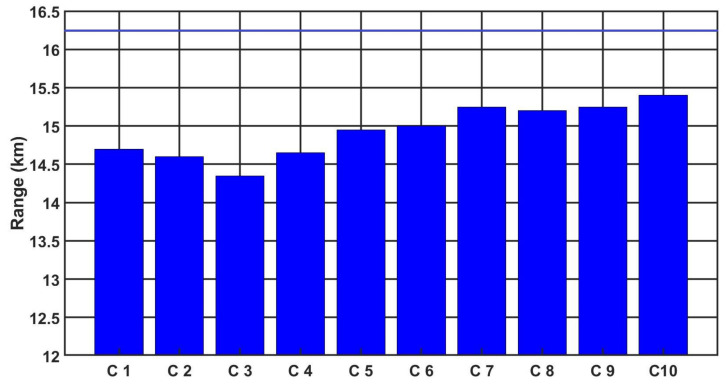
Detection range of the LRIRCs (target: 0.6 m × 1.6 m, ΔT = 2 K). The blue column presents the measured range D for the cameras; the blue line presents the producer declaration of the range D.

**Figure 13 sensors-21-05700-f013:**
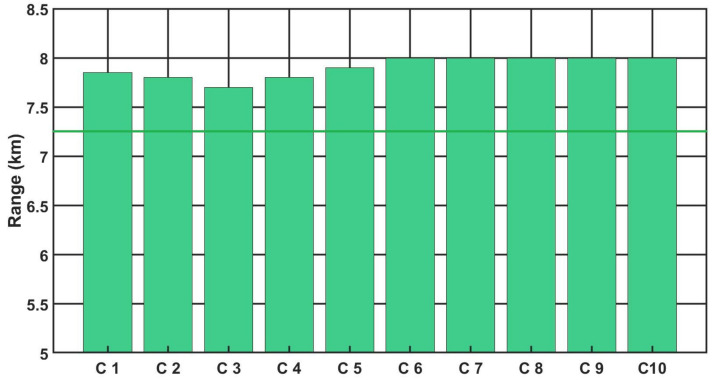
Recognition range of the LRIRCs (target: 0.6 m × 1.6 m, ΔT = 2 K). The green column presents the measured range *R* for the cameras; the green line presents the producer declaration of the range R.

**Figure 14 sensors-21-05700-f014:**
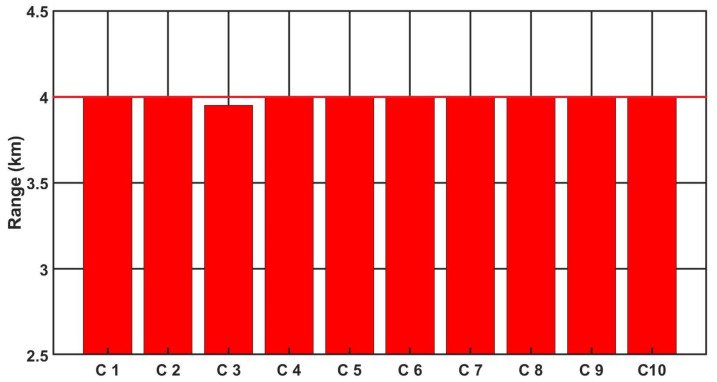
Identification range of the LRIRCs (target: 0.6 m × 1.6 m, ΔT = 2 K). The red column presents the measured range I for the cameras; the red line presents the producer declaration of the range I.

**Figure 15 sensors-21-05700-f015:**
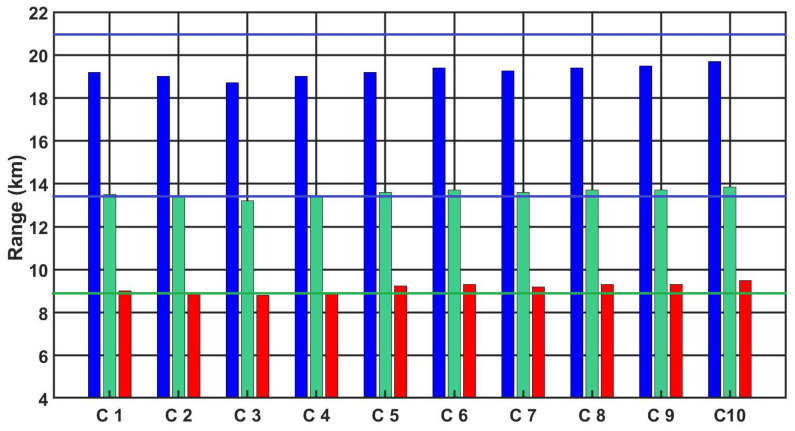
DRI range of the LRIRCs (target: standard target NATO 2.3 m × 2.3 m, ΔT = 2 K). Blue, green and red columns present the measured DRI range for the cameras; blue, green and red lines present the producer declaration of the range DRI.

**Figure 16 sensors-21-05700-f016:**
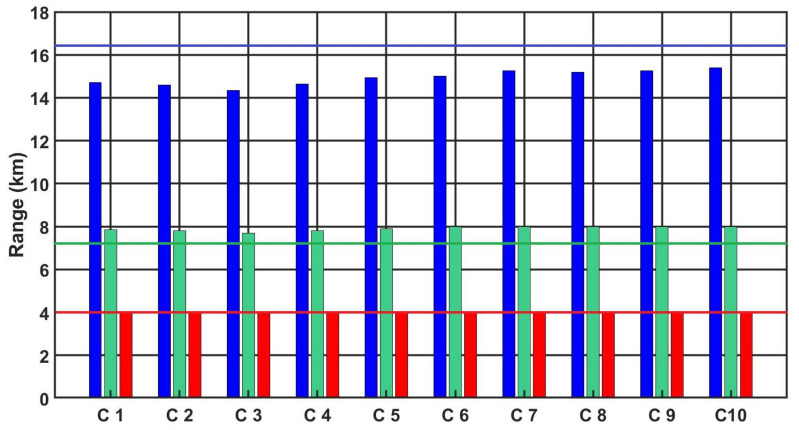
DRI range of the LRIRCs (target: 1.6 m × 0.6 m, ΔT = 2 K). Blue, green and red columns present the measured DRI range for the cameras; blue, green and red lines present the producer declaration of the range DRI.

**Table 1 sensors-21-05700-t001:** Specification of the test stand.

The Test Stand			
METS S-12 IR collimator		SR800R-4D-HE blackbody	
Aperture	300 mm	Aperture	4${}^{\prime \prime }\times 4^{\prime \prime }$
EFL	1787 mm	Differential temp. range	−30–100 ${}^{\circ }$C
FOV	1.6	Emissivity	0.98
Transmission	0.98	Uniformity	0.01

**Table 2 sensors-21-05700-t002:** Specification of the camera parameters.

Parameter	Value
Sensor type	InSb
Spectral range	3–5 (μm)
FPA size (H × V)	640 × 480
Pixel pitch	15 (μm)
NETD	<20 mK
Frame rate	50 (Hz)
Field of view (FOV)	0.75${}^{\circ }$ (H) × 0.56${}^{\circ }$ (V)

**Table 3 sensors-21-05700-t003:** Example of the uncertainty analysis for the NETD camera C1.

	NETD1	NETD2	NETD3	NETD4	NETD5
	**20**	**19**	**20**	**21**	**19**
$\bar{NETD}$	19.8				
$\delta S_{max}$	2.34				
$\delta T_{max}$	0.012				
$S_{1}$	895				
$S_{2}$	775				
$S_{n}$	895				
$u_{a}$	0.75				
$u_{b}$	0.44				
u(NETD)	1.18				
U(NETD)	2.37				

**Table 4 sensors-21-05700-t004:** Results of the NETD measurements of 10 cameras.

	NETD1	NETD2	NETD3	NETD4	NETD5	$\bar{\mathrm{NETD}}$
C1	20	19	20	21	19	19.8
C2	19	19	20	20	19	19.4
C3	19	19	20	20	19	19.4
C4	21	21	20	21	21	20.8
C5	20	20	20	19	21	20.0
C6	18	18	17	18	19	18.0
C7	17	18	17	18	19	17.8
C8	18	18	18	18	19	18.2
C9	16	17	17	17	17	16.8
C10	19	19	19	19	19	19.0

**Table 5 sensors-21-05700-t005:** Results of the measurement of the MRTD camera C1 for 3 observers (Obs1, Obs2, Obs3).

γ	Obs1	Obs1	Obs2	Obs2	Obs3	Obs3	Obs1	Obs2	Obs3	MRTD (γ)
	$\Delta T_{-}$	$\Delta T_{+}$	$\Delta T_{-}$	$\Delta T_{+}$	$\Delta T_{-}$	$\Delta T_{+}$	MRTD (γ)	MRTD (γ)	MRTD (γ)	
1.11	0.013	0.013	0.014	0.014	0.012	0.013	0.013	0.014	0.013	0.013
1.27	0.012	0.013	0.014	0.014	0.013	0.013	0.013	0.014	0.013	0.013
1.78	0.013	0.014	0.015	0.016	0.014	0.015	0.014	0.016	0.015	0.015
2.16	0.014	0.015	0.017	0.018	0.015	0.017	0.015	0.018	0.016	0.016
3.92	0.018	0.020	0.020	0.022	0.018	0.018	0.019	0.021	0.018	0.019
5.66	0.027	0.028	0.030	0.030	0.025	0.027	0.028	0.030	0.026	0.028
7.45	0.035	0.037	0.038	0.040	0.035	0.038	0.036	0.039	0.037	0.037
9.21	0.048	0.048	0.050	0.052	0.046	0.046	0.048	0.051	0.046	0.048
10.9	0.057	0.060	0.060	0.065	0.055	0.054	0.059	0.063	0.055	0.059
12.78	0.077	0.082	0.080	0.085	0.074	0.077	0.080	0.083	0.076	0.079
14.41	0.094	0.097	0.100	0.100	0.090	0.090	0.096	0.100	0.090	0.095
16.25	0.105	0.107	0.110	0.115	0.110	0.110	0.106	0.113	0.110	0.110
16.86	0.135	0.134	0.140	0.145	0.135	0.137	0.135	0.143	0.136	0.138
17.87	0.150	0.148	0.155	0.165	0.155	0.152	0.149	0.160	0.154	0.154
19.86	0.197	0.194	0.205	0.195	0.190	0.185	0.196	0.200	0.188	0.194
21.27	0.225	0.220	0.235	0.235	0.225	0.220	0.223	0.235	0.22 3	0.227
23.27	0.292	0.282	0.300	0.285	0.275	0.270	0.287	0.293	0.27 3	0.284
24.82	0.425	0.420	0.440	0.445	0.420	0.415	0.423	0.443	0.418	0.428
25.00	-	-	-	-	-	-	-	-	-	-

**Table 6 sensors-21-05700-t006:** Example of the uncertainty analysis for the MRTD C1.

γ	${\mathrm{MRTD}}_{\mathrm{cor}}\;(\gamma )$	$u\;(\Delta T_{\mathrm{BBA}})$	$u\;(\Delta T_{\mathrm{BBD}})$	$u\;(\Delta T_{\mathrm{BBU}})$	$u\;({\mathrm{MRTD}}_{\mathrm{obs}})$	u(MRTD)
1.11	0.0130	0.0007	0.0013	0.0033	0.0004	0.0036
1.27	0.0130	0.0007	0.0013	0.0033	0.0004	0.0036
1.78	0.0144	0.0007	0.0013	0.0033	0.0006	0.0037
2.16	0.0158	0.0007	0.0013	0.0033	0.0009	0.0037
3.92	0.0191	0.0007	0.0013	0.0033	0.0009	0.0037
5.66	0.0276	0.0007	0.0013	0.0033	0.0012	0.0038
7.45	0.0368	0.0007	0.0013	0.0033	0.0009	0.0037
9.21	0.0479	0.0007	0.0013	0.0033	0.0015	0.0039
10.9	0.0579	0.0007	0.0013	0.0033	0.0023	0.0043
12.78	0.0784	0.0007	0.0013	0.0033	0.0020	0.0041
14.41	0.0942	0.0007	0.0013	0.0033	0.0029	0.0046
16.25	0.1084	0.0007	0.0013	0.0033	0.0019	0.0041
16.86	0.1363	0.0007	0.0013	0.0033	0.0025	0.0044
17.87	0.1526	0.0007	0.0013	0.0033	0.0032	0.0048
19.86	0.1924	0.0007	0.0013	0.0033	0.0037	0.0051
21.27	0.2244	0.0007	0.0013	0.0033	0.0042	0.0055
23.27	0.2812	0.0007	0.0013	0.0033	0.0060	0.0070
24.82	0.4232	0.0007	0.0013	0.0033	0.0076	0.0084
25.00	-	-	-	-	-	-

**Table 7 sensors-21-05700-t007:** The results of an MRTD calculation for 10 cameras.

γ	C1	C2	C3	C4	C5	C6	C7	C8	C9	C10
1.11	0.013	0.013	0.014	0.015	0.014	0.014	0.012	0.011	0.011	0.011
1.27	0.013	0.013	0.015	0.015	0.014	0.014	0.013	0.012	0.012	0.012
1.78	0.014	0.015	0.016	0.016	0.015	0.015	0.013	0.012	0.012	0.013
2.16	0.016	0.017	0.017	0.017	0.016	0.015	0.015	0.013	0.014	0.013
3.92	0.019	0.021	0.023	0.019	0.018	0.021	0.018	0.017	0.017	0.016
5.66	0.028	0.028	0.031	0.029	0.026	0.030	0.027	0.026	0.025	0.024
7.45	0.037	0.037	0.041	0.040	0.036	0.039	0.035	0.036	0.034	0.032
9.21	0.048	0.051	0.055	0.050	0.047	0.052	0.048	0.045	0.045	0.043
10.9	0.058	0.062	0.070	0.062	0.058	0.063	0.062	0.054	0.054	0.055
12.78	0.078	0.080	0.086	0.078	0.074	0.073	0.074	0.069	0.066	0.064
14.41	0.094	0.097	0.105	0.093	0.089	0.087	0.086	0.085	0.084	0.081
16.25	0.108	0.111	0.122	0.106	0.103	0.101	0.098	0.096	0.099	0.094
16.86	0.136	0.136	0.138	0.121	0.119	0.111	0.115	0.108	0.119	0.110
17.87	0.153	0.160	0.167	0.141	0.139	0.136	0.140	0.133	0.139	0.132
19.86	0.192	0.196	0.206	0.191	0.183	0.175	0.177	0.175	0.172	0.161
21.27	0.224	0.233	0.244	0.236	0.227	0.213	0.222	0.209	0.216	0.202
23.27	0.281	0.296	0.314	0.288	0.287	0.254	0.299	0.250	0.256	0.240
24.82	0.423	0.435	0.446	0.436	0.422	0.347	0.366	0.347	0.359	0.293
25.00	-	-	-	0.693	0.693	0.426	0.449	0.411	-	0.369

**Table 8 sensors-21-05700-t008:** Results of the calculation of the DRI parameter (target: standard target NATO 2.3 m × 2.3 m, ΔT = 2 K) for 10 tested LRIRCs.

	Detection	Recognition	Identification
C1	19,200	13,500	9000
C2	19,000	13,400	8900
C3	18,700	13,200	8800
C4	19,000	13,450	8950
C5	19,400	13,700	9300
C6	19,200	13,600	9250
C7	19,250	13,600	9200
C8	19,400	13,700	9300
C9	19,300	13,500	9200
C10	19,700	13,850	9500
Producer	20,800	12,800	8900

**Table 9 sensors-21-05700-t009:** Results of the calculation of the DRI parameter (target: man 0.6 m × 1.6 m, ΔT = 2 K) for 10 tested LRIRCs.

	Detection	Recognition	Identification
C1	14,700	7850	4000
C2	14,600	7800	4000
C3	14,350	7700	3950
C4	14,650	7800	4000
C5	14,950	7900	4000
C6	15,000	8000	4000
C7	15,250	8000	4000
C8	15,200	8000	4000
C9	15,250	8000	4000
C10	15,400	8000	4000
Producer	16,300	7200	4000

**Table 10 sensors-21-05700-t010:** Comparison of the DRI parameters for camera C3 and C10.

	Target NATO			Target Man		
	**Camera C3**	**Camera C10**	**Difference**	**Camera C3**	**Camera C10**	**Difference**
D	18,700	19,700	1000	14,350	15,400	950
R	13,200	13,850	650	7700	8000	300
I	8800	9500	700	3950	4000	50

**Table 11 sensors-21-05700-t011:** Comparison of the detection range D for 10 cameras and the producer declaration.

	Target NATO			Target Man		
**Camera**	**Measured**	**Producer**	**Difference**	**Measured**	**Producer**	**Difference**
C1	19,200	20,800	1600	14,700	16,300	1600
C2	19,000	20,800	1800	14,600	16,300	1700
C3	18,700	20,800	2100	14,350	16,300	1950
C4	19,000	20,800	1800	14,650	16,300	1650
C5	19,400	20,800	1400	14,950	16,300	1350
C6	19,200	20,800	1600	15,000	16,300	1300
C7	19,250	20,800	1550	15,250	16,300	1050
C8	19,400	20,800	1400	15,200	16,300	1100
C9	19,300	20,800	1500	15,250	16,300	1150
C10	19,700	20,800	1100	15,400	16,300	900

## Data Availability

The data presented in this study are available on request from the corresponding author.

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
