# Peer review of "Measurement and Analysis of the Parameters of Modern Long-Range Thermal Imaging Cameras"

_sensors, 2021, doi:10.3390/s21175700_

Round 1

Reviewer 1 Report

The paper presents the results of Measurement and analysis of the parameters of modern long range thermal imaging cameras. The topic is interesting and new. The introduction is poor, as it is focused only in the specific long range cameras, without showing the gap with the existing literature. The references are poor. Review papers on irt of Balaras, Nardi, Lucchi, and Tejedor are particularly useful to enrich this part. The structure of the paper is not clear. Figure 3 is very helpful for understanding the parameters used in irt. You wrote that only references 14-17 are able to do irt tests. Where do you find this information? Add these details. The discussion on parameters is well argumented. Thank n the contrary, the part on the results is not understandable at all. Explicate better the results, adding some sentences as the table and the graphics must be interpreted. Conclusion must be revised totally. English need a revision from a mothertongue.

Round 2

Reviewer 1 Report

English correction is needed